# Experimental Investigation of the Optical Nonlinearity of Laser-Ablated Titanium Dioxide Nanoparticles Using Femtosecond Laser Light Pulses

**DOI:** 10.3390/nano14231940

**Published:** 2024-12-03

**Authors:** Fatma Abdel Samad, Mohammed Ali Jasim, Alaa Mahmoud, Yasmin Abd El-Salam, Hamza Qayyum, Retna Apsari, Tarek Mohamed

**Affiliations:** 1Laser Institute for Research and Applications LIRA, Beni-Suef University, Beni-Suef 62511, Egypt; fatmaabdalsamad@lira.bsu.edu.eg (F.A.S.); alaa.abutaleb@lira.bsu.edu.eg (A.M.); yasmena@lira.bsu.edu.eg (Y.A.E.-S.); 2Anbar Health Department, Ministry of Health, Ramadi 31001, Iraq; alijasim1980@lira.bsu.edu.eg; 3Laser-Matter Interaction Laboratory, Department of Physics, COMSATS University Islamabad, Park Road, Islamabad 45550, Pakistan; 4Department of Engineering, Faculty of Advanced Technology and Multidiscipline, Universitas Airlangga, Surabaya 60115, Indonesia; apsari.unair@gmail.com

**Keywords:** laser ablation, nonlinear optics, Z-scan approach, nonlinear absorption coefficient, titanium dioxide, nonlinear refractive index, femtosecond laser, optical limiting effect

## Abstract

In this report, the nonlinear optical (NLO) properties of titanium dioxide nanoparticles (TiO_2_ NPs) have been explored experimentally using femtosecond laser light along with the Z-scan approach. The synthesis of TiO_2_ NPs was carried out in distilled water through nanosecond second harmonic Nd:YAG laser ablation. Characterization of the TiO_2_ NPs colloids was conducted using UV-visible absorption spectroscopy, transmission electron microscopy (TEM), inductively coupled plasma (ICP), and energy-dispersive X-ray spectroscopy (EDX). The TEM analysis indicated that the size distribution and average particle size of the TiO_2_ NPs varied from 8.3 nm to 19.1 nm, depending on the laser ablation duration. The third-order NLO properties of the synthesized TiO_2_ NPs were examined at different excitation laser wavelengths and incident powers through both open- and closed-aperture Z-scan techniques, utilizing a laser pulse duration of 100 fs and a high repetition rate of 80 MHz. The nonlinear absorption (NLA) coefficient and nonlinear refractive (NLR) index of the TiO_2_ NPs colloidal solutions were found to be influenced by the incident power, excitation wavelength, average size, and concentration of TiO_2_ NPs. Maximum values of 4.93 × 10⁻⁹ cm/W for the NLA coefficient and 15.39 × 10⁻^15^ cm^2^/W for the NLR index were observed at an excitation wavelength of 800 nm, an incident power of 0.6 W, and an ablation time of 15 min. The optical limiting (OL) effects of the TiO_2_ NPs solution at different ablation times were investigated and revealed to be concentration and average size dependent. An increase in concentration results in a more limiting effect.

## 1. Introduction

Optical materials that exhibit low absorption and high optical nonlinearity are often considered ideal for photonic and optoelectronic technologies [1]. The exploration of nanoparticles (NPs) in optical applications has captured the attention of researchers due to their unique properties [2]. These nanostructures, with their sub-nano dimensions and exotic shapes, display significant quantum confinement effects, making them suitable for a broad array of applications, including photonics, solar cells, photocatalysis, catalytic chemistry [3], and nonlinear optics [4,5,6].

Nanostructured semiconductors, in particular, possess notable linear and nonlinear optical (NLO) characteristics, which makes them valuable in fields such as photonics, gas sensing, optoelectronics, field emission displays, and medical technology [7,8,9]. Among various nanostructured semiconductors, titanium dioxide (TiO_2_) NPs hold significant potential in diverse applications due to their wide band gap of approximately 3.2 eV [10]. Nanocrystalline TiO_2_ is one of the most thoroughly researched materials due to its unique physical and chemical characteristics. It finds applications in solar and fuel cells [11] and protective and self-cleaning coatings, catalysis, and photocatalysis [12,13]. TiO_2_ NPs are especially prominent in optoelectronic applications, sparking considerable interest within the research community [14,15,16]. When exposed to intense collimated laser beams, the optically induced properties of TiO_2_ NPs exhibit notable NLO characteristics. The linear and NLO responses of bulk TiO_2_ materials within their transparency range can be explained by virtual transitions between the filled O 2p valence band and the vacant Ti 3d cationic band [17]. Numerous methods are employed to study the NLO properties of materials [18,19,20,21,22], though many are hindered by inaccuracies, complexity, and the need for intricate wave propagation analysis [23]. Among these, the Z-scan technique stands out as the simplest and most effective method for measuring the nonlinear refractive (NLR) index and nonlinear absorption (NLA) coefficient of various materials [24,25,26,27,28]. Open aperture (OA) and closed aperture (CA) Z-scan techniques are commonly used in experiments to determine the NLA coefficient and NLR index [29,30,31]. Two-photon absorption (2PA) is the most prevalent NLA phenomenon [32], where two photons are absorbed simultaneously, transitioning from a lower to a higher energy state.

Various techniques are available for preparing nanoparticle colloids, but pulsed laser ablation in liquid (PLAL) offers a promising alternative to traditional chemical methods. PLAL enables the synthesis of stable metal colloids in pure solvents without the need for toxic or hazardous chemicals [33]. By adjusting laser parameters such as pulse duration, fluence, repetition rate, wavelength, and pulse width, the size of nanoparticles can be precisely controlled, making this method highly versatile and effective. Numerous studies have explored the NLO properties of TiO_2_ in various forms, including NPs, thin films, nanocomposites, and nanocolloidal solutions [34,35]. However, only a few have examined the NLO behavior of TiO_2_ nanocolloids in liquid form, particularly those synthesized using the laser ablation method [36,37]. For example, TiO_2_ NPs were produced via laser ablation by irradiating a titanium target in distilled water at 1064 nm [36]. The Z-scan technique, using a continuous-wave He–Ne laser at different power levels, was employed to investigate the NLO properties of these TiO_2_ colloids, revealing a negative NLR index (n_2_) and a positive NLA coefficient (β) [36]. In another study [37], TiO_2_ NPs were synthesized by femtosecond (fs) laser ablation at 800 nm in deionized water, and the NLO properties were analyzed using a 150 fs laser across wavelengths of 700 nm, 750 nm, 800 nm, and 850 nm. The results showed reverse saturable absorption (RSA) and negative n_2_ values for the TiO_2_ nanocolloids [37].The aim of this study is to explore the NLO properties of TiO_2_ nanocolloidal solutions synthesized via laser ablation in distilled water, without the use of chemical additives. Nanosecond (ns) laser pulses are employed to generate TiO_2_ NPs under varying laser conditions. The synthesized TiO_2_ NPs are characterized using techniques such as UV-Vis spectroscopy, energy dispersive X-ray (EDX), transmission electron microscopy (TEM), and inductively coupled plasma (ICP). Femtosecond (fs) laser pulses are then used to study the NLA coefficient and NLR index of TiO_2_ colloids at different excitation wavelengths and power levels, utilizing OA and CA Z-scan methods. Additionally, the study examines the impact of various TiO_2_ nanoparticle concentrations and particle sizes on the NLO susceptibility and optical limiting behavior.

## 2. Experimental Setup

### 2.1. TiO_2_ Sample Preparation

Figure 1 illustrates the experimental setup for synthesizing TiO_2_ nanoparticle colloids through laser ablation, using a second harmonic Nd:YAG laser system (Spectra Physics-Quanta-Ray PRO 350, (Milpitas, CA, USA)) with a pulse duration of 10 ns and a repetition rate of 10 Hz. The 532 nm laser operates with a maximum pulse energy of 1500 mJ per pulse, and the beam has a Gaussian distribution with a TEM00 mode. A titanium sample (purity > 99%) was submerged in distilled water to generate the TiO_2_ colloids. A convex lens with a 10.5 cm focal length was used to focus the laser beam, with an average power of 150 mW and energy of 15 mJ per pulse. The focal spot size, determined via the knife-edge method, was around 1.75 mm. The titanium sample was immersed in 10 mL of distilled water inside a beaker. To ensure uniform distribution and prevent particle interference during ablation, the beaker was rotated at 177 RPM using a motorized spinner. TiO_2_ NPs were synthesized at varying ablation times of 5, 10, and 15 min.

### 2.2. Z-Scan Setup

The NLO properties of TiO_2_ NPs were analyzed using a Z-scan setup, as shown in Figure 2 [28,38]. This experiment employed an fs laser system (INSPIRE HF100, Spectra-Physics, (Milpitas, CA, USA)), pumped by a mode-locked fs Ti:Saphire laser (MAI TAI HP, Spectra-Physics, (Milpitas, CA, USA)), which offers tunable wavelengths from 690 to 1040 nm, an average power output of 1.5–2.9 W, and an 80 MHz repetition rate. The Inspire laser system includes four output apertures, with the first delivering fundamental infrared pump wavelengths, while the others are driven by two additional modes based on NLO fields: a second harmonic generator (SHG) and an optical parametric oscillator (OPO). These modes allow for output wavelengths spanning from 345 nm to 2500 nm. To examine the NLO properties of TiO_2_ colloidal solutions, 100 fs laser pulses with Gaussian distribution were applied at various excitation wavelengths ranging from 750 nm to 850 nm. The laser beam, characterized by a Gaussian spatial profile (TEM00) and an M^2^ value of less than 1.1, was tightly focused using a 5 cm focal length convex lens. The TiO_2_ colloidal solution was contained in a 1 mm path-length quartz cuvette, mounted on a micrometer translation stage to enable scanning around the focal point. The transmitted intensity of the samples was recorded using a power meter (PM, Newport 843 R) as a function of their position relative to the focus.

For the CA Z-scan measurements, the aperture was set to S = 0.3, and the transmitted intensity was recorded by PM1, allowing for the determination of both the sign and magnitude of the NLR index. In the OA Z-scan measurements, where S = 1 (fully open), the NLA coefficient of the TiO_2_ samples was obtained, with the transmitted intensity recorded by PM2.

The experimental error in the obtained NLO coefficients was approximately 10%, which mainly originated from the determination of the irradiance distribution utilized in the experiment, i.e., beam waist, pulse width, and laser power calibration.

## 3. Results and Discussion

### 3.1. TiO_2_ NP Colloidal Solution Characterization

The optical absorption of the TiO_2_ NP colloids was measured using UV-visible absorption spectroscopy (Peak Instruments C-7200, Inchinnan, UK) across a wavelength range of 200 to 1100 nm. Figure 3 shows the UV-Vis absorption spectra for TiO_2_ NP colloids formed at different ablation times of 5, 10, and 15 min. The concentrations of the TiO_2_ NP colloids were measured using an ICP device (Agilent 5100 Synchronous Vertical Dual View (SVDV) ICP-OES, Agilent Vapor Generation Accessory VGA 77, Merck Company (Darmstadt, Germany)). The results indicated that the colloids synthesized for 5, 10, and 15 min had concentrations of 1.9 mg/L, 2.9 mg/L, and 4.35 mg/L, respectively. Figure 3 shows how the concentration and average size of TiO_2_ NPs significantly affect the absorbance spectrum. A surface plasmon resonance (SPR) peak is observed around 227 nm for the colloidal TiO_2_ NPs, with the concentration and average size of the TiO_2_ NPs influencing the SPR peak’s position. The absorption peak maxima occur at 228.5 nm, 227 nm, and 225.3 nm for ablation times of 5 min, 10 min, and 15 min, respectively. This absorption peak corresponds to the excitation of electrons from the valence band to the conduction band in TiO_2_ NPs. As the laser ablation time increases, a shift of the peak toward shorter wavelengths is observed, indicating the formation of smaller particles. Additionally, longer ablation times result in sharper SPR peaks, particularly at 15 min of ablation.

In the visible region, the TiO_2_ NP samples show high optical transparency, a crucial factor for optoelectronic applications. It was also noted that increasing the ablation time led to higher absorbance in this region. The energy band gap (E_g_) of the TiO_2_ NP colloidal solutions is calculated from the optical absorption data using Tauc’s plot equation [39].
(1)$${\left(\alpha h\upsilon \right)}^{\frac{1}{2}}=a(h\upsilon -E_{g})$$
where a is a constant, and α is the linear absorption coefficient, which is obtained via the relation $\alpha =\frac{2.303A}{t}$ [40], where A is the absorbance, and t is the thickness of the sample. The absorption spectrum revealed that the value of α varied with the ablation time, as shown in Figure 3. Figure 4a–c depicts the E_g_ values for the TiO_2_ NP solutions obtained from the (αhυ)^1/2^ versus hυ plots [41]. The E_g_ values of the TiO_2_ NP colloids can be deduced by extrapolating the linear part of the plot to αhυ = 0. As shown in Figure 4, the E_g_ of TiO_2_ NP colloidal solutions depends on the sample concentration and average size of NPs and decreases from 3.06 eV to 2.84 eV as the ablation time increases from 5 min to 15 min. The transmission results were used to calculate the linear refractive index (n_0_) via the Swanepoel formalism [42].
(2)$$n_{0}=\frac{1}{T}+{\left(\frac{1}{T^{2}}-1\right)}^{1/2}$$
where T is the transmittance of the TiO_2_ NP colloidal solution. The linear refractive index was calculated for each concentration of TiO_2_ NPs and is summarized in Table 1.

The size distribution and average size of TiO_2_ NPs were determined using high-resolution transmission electron microscopy (HR-TEM, JEM-2100, Joel, Japan, operated at 200 kV) at various ablation times. Figure 5a–c presents the size distribution histograms and TEM images of the TiO_2_ NP colloids for ablation times of 5, 10, and 15 min, corresponding to concentrations of 1.9 mg/L, 2.9 mg/L, and 4.35 mg/L, respectively. At a laser fluence of 1.24 J/cm^2^, the TiO_2_ nanoparticle colloids displayed a spherical morphology. For consistency, all images were captured at the same magnification, and particle sizes were analyzed using calibrated imaging software (Image J 1.45). The average size of TiO_2_ NPs was determined by taking four images for each ablation time and at different sample positions. The average sizes of the TiO_2_ NP samples were found to be 19.11 nm, 11.96 nm, and 8.33 nm for the ablation times of 5, 10, and 15 min, respectively. The decrease in average particle size suggests that the photo-fragmentation process becomes more efficient as the ablation duration increases.

The elemental composition of the TiO_2_ NP samples was analyzed using EDX with a JSM6510LA Eds detector from Oxford Instruments (Halifax Road, High Wycombe Buckinghamshire, UK). The EDX measurements, performed to determine the atomic composition of the TiO_2_ colloidal solutions, were conducted with a 20 kV electron beam and a spectrum counting time of 35 s. Figure 6 presents the EDX spectra of the TiO_2_ colloidal solution. The ZAF method, which accounts for atomic number (Z), absorption (A), and fluorescence (F), was applied to determine the elemental composition of the TiO_2_ NPs. The spectrum clearly shows the presence of titanium (Ti) and oxygen (O), with atomic percentages of 31.51% and 68.49%, respectively, as summarized in Figure 6.

### 3.2. Open Aperture Z-Scan Measurements

#### 3.2.1. The Effect of Incident Power on β

Figure 7 shows the experimental OA Z-scan of TiO_2_ NPs investigated using a high repetition rate (HRR, 80 MHz) fs laser. The NLO properties of TiO_2_ NPs at the different ablation times were studied at various incident powers ranging from 0.6 to 1.2 W at an 800 nm excitation wavelength. Figure 7a–c depicts that all curves represent RSA. The absorption effect depends on the laser intensity, and normalized transmittance is symmetric around the focus (Z = 0), with the lowest transmission at the focus (valley) [25,43]. The NLA coefficient depends on the laser intensity (I) delivered to the sample when exposed to high laser intensity, as shown by the following equation [29,30]:(3)$$\propto \left(I\right)=\alpha +\beta I$$
where ꞵ is the 2PA coefficient. To obtain the NLA coefficient, the OA Z-scan measurements were theoretically fitted using the NLA model provided by [38,44,45]
(4)$$T_{OA}=1-\left.\left(\frac{\beta {I_{0}}^{\left(n-1\right)}L_{eff}}{{\left(m+1\right)}^{\frac{3}{2}}{\left(1+\frac{Z^{2}}{{Z_{0}}^{2}}\right)}^{\left(n-1\right)}}\right.\right)$$
where I_0_ is the peak intensity at the focus (Z = 0), and m = 1 for 2PA, and m = 2 for three-photon absorption (3PA). Z_0_ is the Rayleigh length $Z_{0}=\frac{n_{0}\pi {\omega_{0}}^{2}}{\lambda }$. The Rayleigh length was greater than the sample thickness ($Z_{0}>L$). $L_{eff}=1-(e^{-m\alpha L}/m\alpha$), where L is the thickness of the sample, $\omega_{0}$ is the beam waist at the focus (16 μm ± 1.6), and λ is the excitation wavelength. Normalized transmittance (T_OA_) decreases as the incident power increases, as illustrated in Figure 7a–c. Figure 7d depicts the dependence of the NLA coefficient on the incident laser power; as the incident power increases, the NLA coefficient decreases. Based on the absorption mechanism of semiconducting materials, NLA can be divided into RSA, multiphoton absorption, excited-state absorption (ESA), and free carrier absorption (FCA). The number of free electrons in the conduction band (Ti 3d) and holes in the valence band (O 2p) increases as the incident power increases. The collisions of electrons with each other in the conduction band are called “many-body interactions” [46]. As a result, collisions between free carriers increase, the scattering of photons and phonons increases, and ꞵ decreases.

#### 3.2.2. Effect of the Excitation Wavelength on β

Figure 8a–c depicts the dependence of the NLA coefficient of TiO_2_ NP colloidal solutions on the excitation wavelength ranging from 750 nm to 850 nm, as well as different ablation times. The T_OA_ decreases as the excitation wavelength increases, showing an increase in RSA [37]. TiO_2_ NPs exhibit an energy band gap from 3.06 eV to 2.84 eV at ablation times from 5 min to 15 min. Two-photon absorption occurs with photon energies ranging from 1.65 to 1.458 eV, which corresponds to the used wavelength range from 750 to 850 nm. Figure 8d depicts the effect of excitation wavelength on the NLA coefficient of TiO_2_ NP solutions at 1 W incident power. Increasing the excitation wavelength stimulates more electrons to fill the excited bands. TiO_2_ NP solutions exhibit a linear increase in the NLA coefficient as wavelength increases. As the laser wavelength decreases, it approaches the far tail of linear absorption (one-photon absorption).

#### 3.2.3. Effect of Ablation Time on β of TiO_2_ NPs

As shown in Figure 9a, the OA Z-scan measurements of the TiO_2_ NP colloidal solutions were studied at different ablation times using an 800 nm excitation wavelength and 1 W incident power. The T_OA_ decreased with increasing ablation time and exhibited a distinct reduction in transmittance when the laser focused on the TiO_2_ NP sample; a typical RSA response was observed. The NLA coefficient of the TiO_2_ NP colloidal solutions is linearly related to the ablation time, as depicted in Figure 9b. As the ablation time increases, TiO_2_ NP concentration increases, and the number of NP molecules involved in the laser interaction increases, which leads to an increase in the number of two-photon absorbers. The decrease in the NLA coefficient with increasing NP size is due to the larger number of NPs that can be accommodated in a given volume as particle size decreases [47,48].

### 3.3. CA Z-Scan Measurements: The Effect of Incident Power on n_2_

The experimental CA Z-scan of TiO_2_ NP colloids was studied via 100 fs HRR laser pulses. The HRR laser has a 12.5 ns separation time between pulses. For liquids and optical glasses, the thermal characteristic time t_c_ =$\frac{\omega^{2}}{4D}$, t_c_ is ≥ 40 μs [49], where D represents the sample’s thermal diffusion coefficient, and ω denotes the incident laser beam waist. The separation time is less than t_c_. Cumulative heating occurs when the sample does not return to its equilibrium temperature between pulses. This heating can lead to a temperature distribution that alters the spatial distribution of the refractive index, resulting in distorted CA Z-scan experimental data and inaccurate NLR index results. For the steady-state case and the thermal nonlinearity, the on-axis change in the NLR index Δn can be given as follows [50]:(5)$$∆n=\frac{dn}{dT}\times \frac{I\alpha {\omega_{o}}^{2}}{4\kappa }$$
where κ represents the thermal conductivity, and $\frac{dn}{dT}$ represents the thermo-optic coefficient. The NLR index of TiO_2_ NP colloidal solutions depends on the excitation laser parameters. The repetition rate, the number of laser pulses that fall on the sample during the scan, and the sample position all have an impact on accumulative thermal lensing, which can be described as follows:(6)$$\frac{1}{f(Z)}=\frac{aLE_{p}{F_{l}}^{\frac{3}{2}}}{{\omega \left(z\right)}^{2}}\left(1-\frac{1}{\sqrt{N_{p}}}\right)$$
where F_l_ is the repetition rate; a represents the fitting parameter a = α (dn/dT)/2κ (π^3^D)^1/2^; $\omega \left(z\right)$ represents the radius of the laser beam at the sample; E_p_ represents the energy per pulse; and N_p_ represents the number of laser pulses incident on the sample. N_p_ = t × F_l_, where t is the time the TiO_2_ NP sample takes to complete the scan. In this work, t is approximately 3 min, and F_l_ = 80 × 10^6^ s^−1^; then, the TiO_2_ NP sample is exposed to N_p_ = 14 × 10^9^ laser pulses during each scan. The normalized transmittance CA Z-scan measurements (ΔT_CA_) depend on the focal length of the induced lensing at f ≥ Z_0_ [51,52].
(7)$${∆T}_{CA}=1+\frac{2Z}{f(Z)}$$

The CA Z-scan experimental results at various incident powers, ablation times, and an excitation wavelength of 800 nm are displayed in Figure 10a–c. By using Equations (6) and (7), the experimental results are simulated via the theoretical fitting expressed as the solid lines. The on-axis nonlinear phase shift ∆φ as a function of the thermal focal length at the focus can be computed using the procedure outlined in the reference [30]. The fitting parameter (a) and value of f(0) were determined from the best simulation of the experimental results.
(8)$$∆\varphi =\frac{Z_{0}}{2f(0)}$$
where f (0) represents the focal length of the induced thermal lens when the sample is placed at focus (Z = 0). The NLO phase shift produces the NLR index (n_2_), which can be expressed as [51]
(9)$$n_{2}=\frac{\lambda {\omega_{0}}^{2}\Delta \varphi }{\left(2P_{p}\times L_{eff}\right)}$$
where P_p_ represents the peak power. Figure 10 shows that the peak-to-valley normalized transmittance difference (ΔT_p−v_) increases as the incident power increases, which is related to the ∆φ, thermal contribution, and electronic effect of the TiO_2_ NPs. At an excitation wavelength of 800 nm, when the incident power is varied from 0.6 W to 1.2 W, ∆φ varies from 0.325 rad to 0.474 rad, from 0.386 rad to 0.569 rad, and from 0.502 rad to 0.623 rad at different ablation times from 5 min to 15 min, respectively.

The NLR index was estimated as a function of incident power from the best fit of the CA experimental data depicted in Figure 10a–c using Equations (6)–(9). Figure 10d shows the effect of incident power on the n_2_ of TiO_2_ NP colloids at different ablation times and an excitation wavelength of 800 nm.

### 3.4. Investigating the Effect of the Excitation Wavelength on the Nonlinear Refractive Index

Figure 11a–c illustrates the influence of various excitation wavelengths, ranging from 750 to 850 nm, on the n_2_ of TiO_2_ NPs at different ablation times and with an incident power of 1 W. The ΔT_p−v_ varies based on the excitation wavelength. Figure 11d shows that n_2_, derived from Figure 11a–c, varies as a function of the excitation wavelength for ablation times of 5 min, 10 min, and 15 min. At an ablation time of 5 min, n_2_ exhibits a slight decrease as the excitation wavelength increases. Conversely, at 10 min, n_2_ shows a slight increase with higher excitation wavelengths. For the ablation time of 15 min, n_2_ increases with rising excitation wavelength, as depicted in Figure 11c. The dependence of n_2_ on the excitation wavelength is influenced by several factors, including the excitation photon energy, thermal contributions, and the band structure of the TiO_2_ NPs.

### 3.5. Effect of Various Ablation Times on the Nonlinear Refractive Index

The effect of ablation time on the NLR index was studied at an excitation wavelength of 800 nm and an incident power of 1 W. The results of the CA Z-scan for different ablation times are shown in Figure 12a. It was observed that the ΔT_p−v_ increases with higher ablation times. Figure 12b illustrates the relationship between the n_2_ and ablation time. The NLR index of the TiO_2_ NPs was affected by both the average size and concentration of the TiO_2_ NP colloids. The NLR index increases with increasing concentrations and decreases with increasing TiO_2_ NP average sizes. This inverse relationship arises because larger NPs result in fewer particles being accommodated in the same volume, leading to a reduction in the volume fraction. Consequently, this reduction results in lower values for both the NLR index and the NLA coefficient [53].

### 3.6. NLO Susceptibility of TiO_2_ NP Colloidal Solutions

Using experimental results of the NLR index n_2_ and NLA coefficient ꞵ, the real $Re\left[\chi^{\left(3\right)}\right]$ and imaginary $Im\left[\chi^{\left(3\right)}\right]$ parts of the third-order nonlinear susceptibility $\chi^{\left(3\right)}$ can be deduced. The nonlinear susceptibility of a material can be expressed as follows [54,55]:(10)$$\chi^{\left(3\right)}=Re\left[\chi^{\left(3\right)}\right]+iIm\left[\chi^{\left(3\right)}\right]$$
where the $Re\left[\chi^{\left(3\right)}\right]$ parameter is related to the NLR index n_2_, and the $Im\left[\chi^{\left(3\right)}\right]$ parameter is related to the NLA coefficient ꞵ. The real nonlinear susceptibility $Re\left[\chi^{\left(3\right)}\right]$ can be written as follows:(11)$$Re\left[\chi^{\left(3\right)}\right]=\frac{n_{0}}{3\pi }n_{2}$$

The imaginary nonlinear susceptibility $Im\left[\chi^{\left(3\right)}\right]$ is given as
(12)$$Im\left[\chi^{\left(3\right)}\right]\left(esu\right)=\left[\frac{10^{-7}c{{\lambda n}_{0}}^{2}}{96\pi^{2}}\right]ꞵ$$
where c is the speed of light. The real and imaginary nonlinear susceptibilities were measured for TiO_2_ NP colloids, as summarized in Table 1. The figure of merit (FOM) can be used to characterize the third-order NLO properties of TiO_2_ NP colloids, which is dependent on linear absorption coefficient *α* and can be given as [56,57]
(13)$$FOM=\left|\frac{Im\left[\chi^{\left(3\right)}\right]\;}{\alpha }\right|$$

The absolute value of nonlinear susceptibility is written as
(14)$$\left|\chi^{\left(3\right)}\right|=\sqrt{\left({Re\left[\chi^{\left(3\right)}\right]}^{2}+{Im\left[\chi^{\left(3\right)}\right]}^{2}\right)}$$

The absolute values of $\chi^{\left(3\right)}$ are summarized in Table 1.

### 3.7. Optical Power Limiting Measurements

As optical technology continues to evolve, there is a growing need to regulate laser beam intensity to ensure the safety of both human eyes and optical sensors [58]. Various filters are employed for the protection of these sensors, with optical limiters playing a crucial role and gaining significant attention in recent studies [59]. The optical power limiting (OPL) effect of TiO_2_ NP colloids at different ablation times was investigated experimentally by varying the input power at an excitation wavelength of 800 nm, as illustrated in Figure 13. The TiO_2_ NP solution was positioned at the focal point of a convex lens (5 cm), and the output power was recorded across a range of input powers. The OPL effect exhibited by the TiO_2_ NP solution is dependent on concentration and average size of NPs; an increase in concentration leads to a stronger limiting effect. The data presented in Figure 13 indicate that the saturation input power diminishes with rising TiO_2_ NP concentration, as summarized in Table 1. Experimental results demonstrate that the OPL of TiO_2_ NP colloids effectively attenuates intense and potentially harmful laser beams, resulting in nonlinear extinction.

Table 2 summarizes the NLO properties of the TiO_2_ NP samples from the current and previous studies [10,34,35,36,37,60,61,62,63]. We can conclude that the NLO properties of our TiO_2_ NP samples were strongly dependent on the nanoparticle synthesis technique and the laser parameters, including repetition rate, pulse duration, wavelength, and peak intensity.

## 4. Conclusions

We investigated the optical nonlinearities of TiO_2_ NPs synthesized through laser ablation in distilled water. The TiO_2_ NPs were produced at various ablation times, maintaining a constant laser energy per pulse of 15 mJ, and characterized using TEM to determine the average nanoparticle size at different concentrations. Nonlinear optical studies were conducted on TiO_2_ NPs with average sizes of 19.11 nm, 11.96 nm, and 8.33 nm, corresponding to concentrations of 1.9 mg/L, 2.9 mg/L, and 4.35 mg/L, respectively, employing the Z-scan technique. OA Z-scan measurements using an fs HRR laser were performed at excitation wavelengths between 750 nm and 850 nm and power levels ranging from 0.6 W to 1.2 W. These experiments revealed RSA behavior with significant 2PA coefficients. The NLA coefficient was observed to depend on both power and wavelength. CA Z-scan results indicated negative nonlinearity at each ablation time. Variations in size and concentration have been identified as potential factors influencing the differences in the NLO properties of TiO_2_ NPs. Further research is necessary, particularly with higher concentrations of TiO_2_ NPs, to draw more definitive conclusions and unlock the advanced capabilities of these tunable optical nonlinearities. Investigation into the TiO_2_ NP solution’s optical limiting effects at various ablation times showed that these effects depended on the concentration and average size of NPs. A more limiting effect is produced with higher concentration.

## Figures and Tables

**Figure 1 nanomaterials-14-01940-f001:**
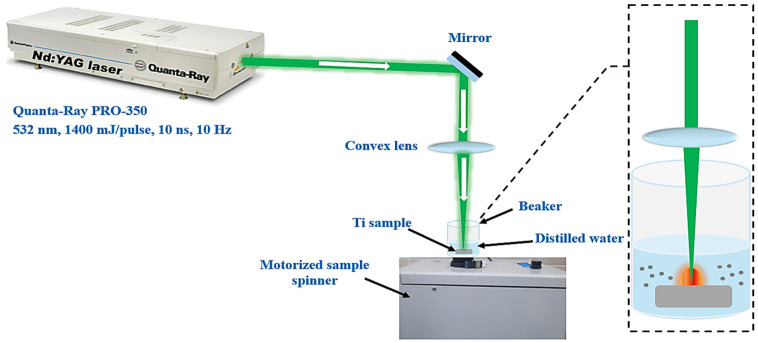
Laser ablation setup for preparing TiO_2_ NPs colloids via a 532-nm Nd:YAG laser.

**Figure 2 nanomaterials-14-01940-f002:**
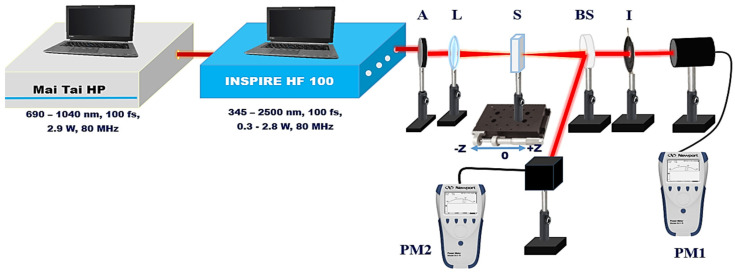
Z-scan experimental setup. L, convex lens; A, attenuator; I, Iris; S, TiO_2_ NPs sample; PM, power meter.

**Figure 3 nanomaterials-14-01940-f003:**
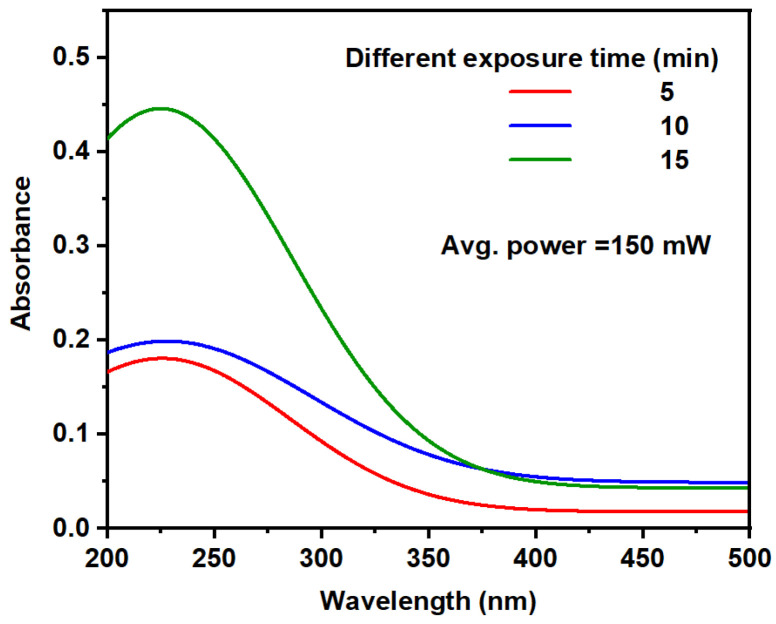
Spectral absorption of TiO_2_ NP colloidal solutions as a function of wavelength at different ablation times.

**Figure 4 nanomaterials-14-01940-f004:**
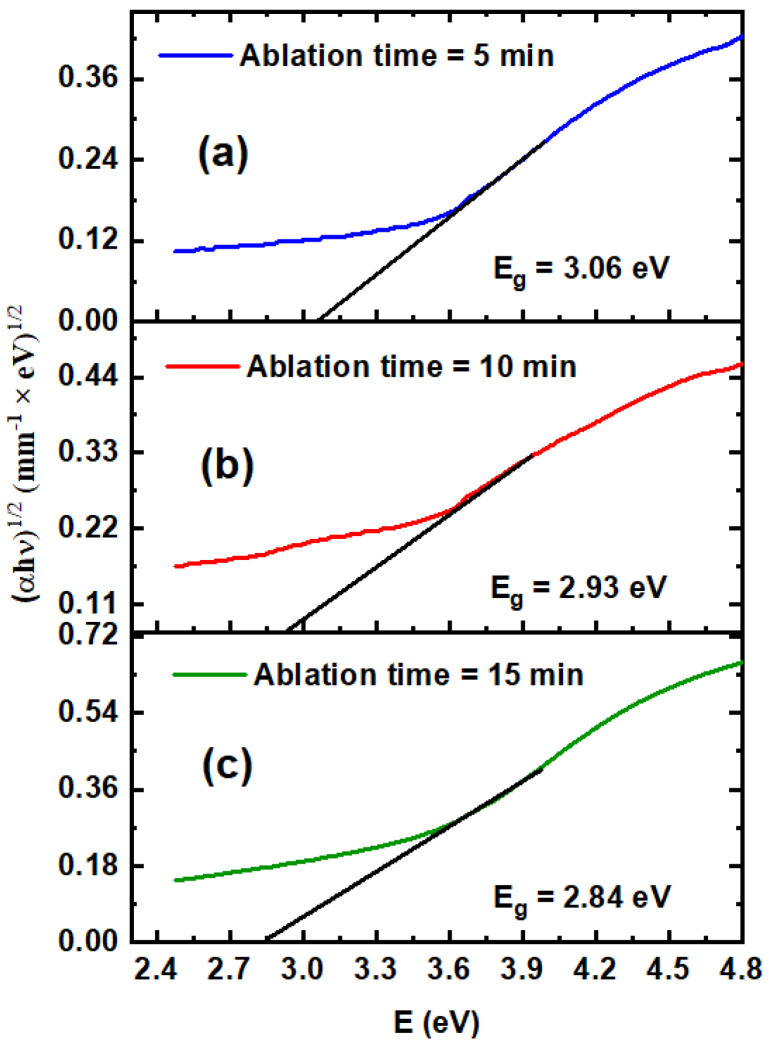
(**a**–**c**) show the energy band gaps that were obtained by extrapolating the straight line of Tauc’s plot of TiO_2_ nanocolloids at various ablation times.

**Figure 5 nanomaterials-14-01940-f005:**
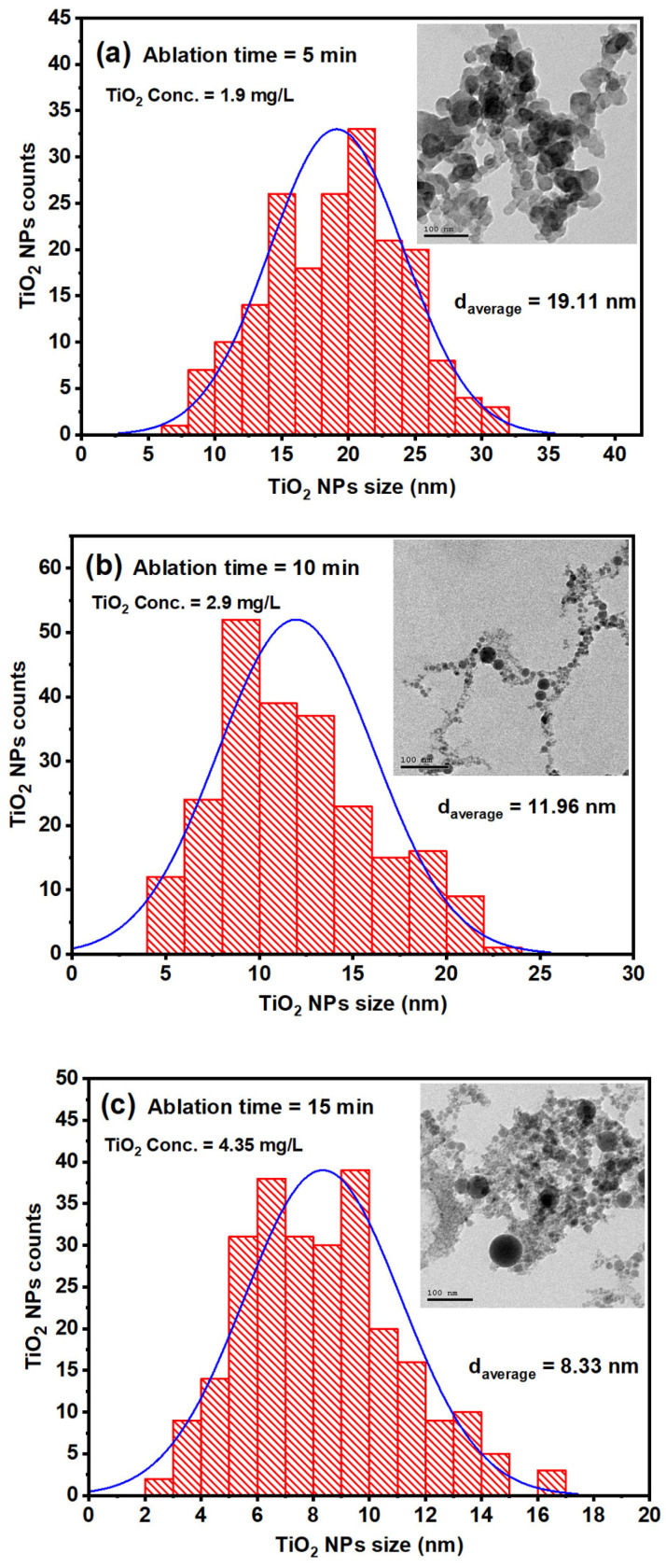
(**a**–**c**) depict the size distributions of TiO_2_ NPs colloids that were synthesized using various ablation times of 5 min, 10 min, and 15 min, respectively.

**Figure 6 nanomaterials-14-01940-f006:**
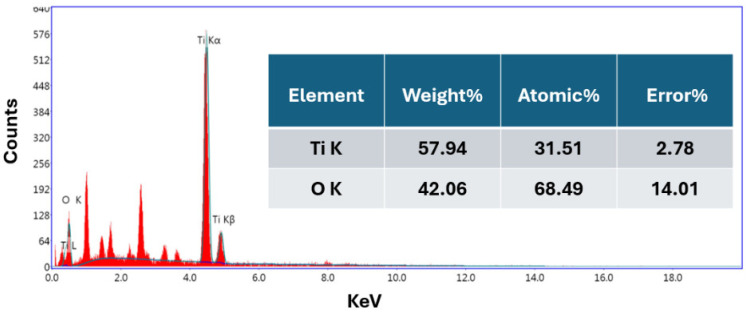
EDX spectra of the TiO_2_ NP colloid and inset ZAF Method Standardless Quantitative Analysis of TiO_2_ NPs.

**Figure 7 nanomaterials-14-01940-f007:**
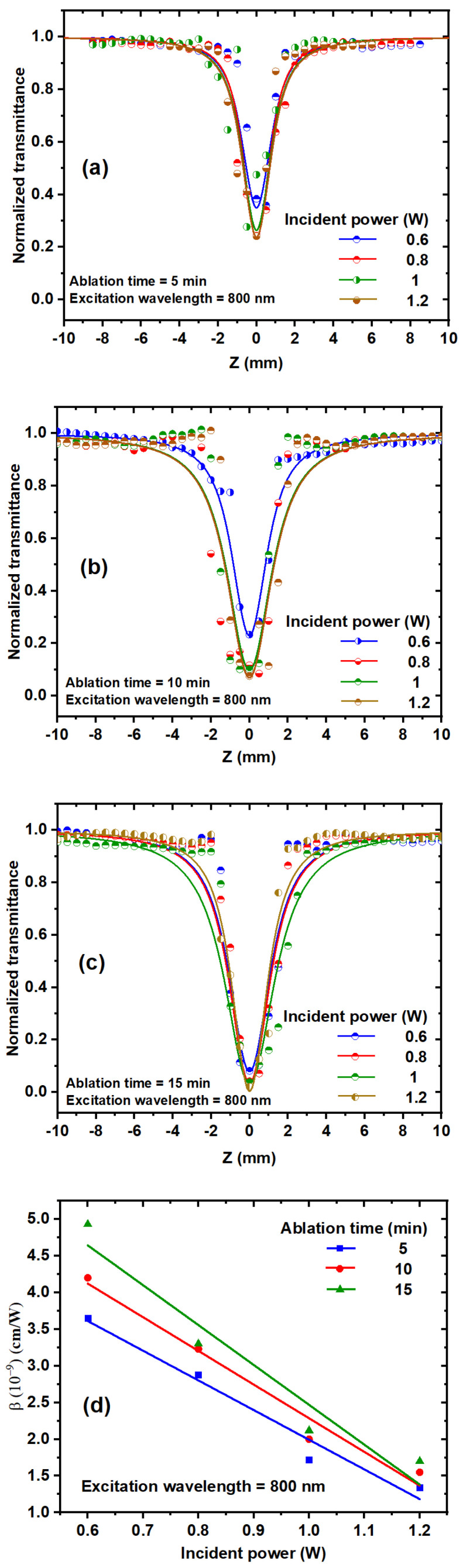
(**a**–**c**) OA Z-scan measurements of TiO_2_ NP colloids with different ablation times and incident powers at an 800 nm excitation wavelength. (**d**) Dependence of the NLA coefficient on the incident laser power at an 800 nm excitation wavelength.

**Figure 8 nanomaterials-14-01940-f008:**
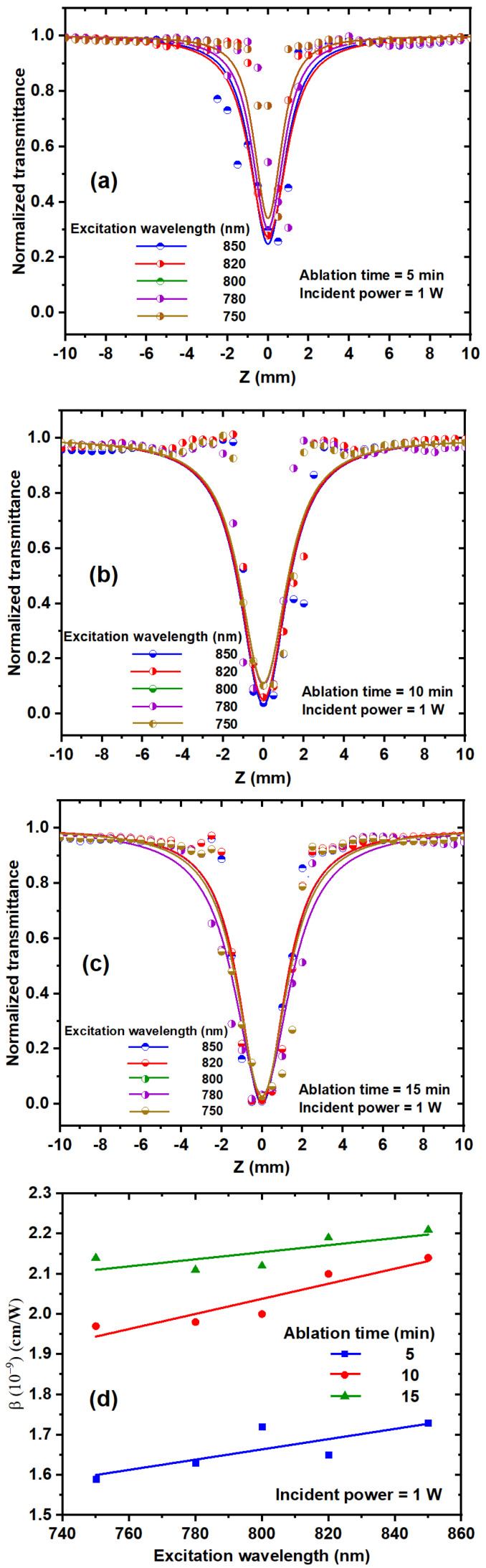
(**a**–**c**) OA Z-scan experimental data of TiO_2_ NP colloidal solutions with different ablation times and excitation wavelengths at 1 W incident power. (**d**) Relationship between the excitation wavelength and NLA coefficient at 1 W incident laser power.

**Figure 9 nanomaterials-14-01940-f009:**
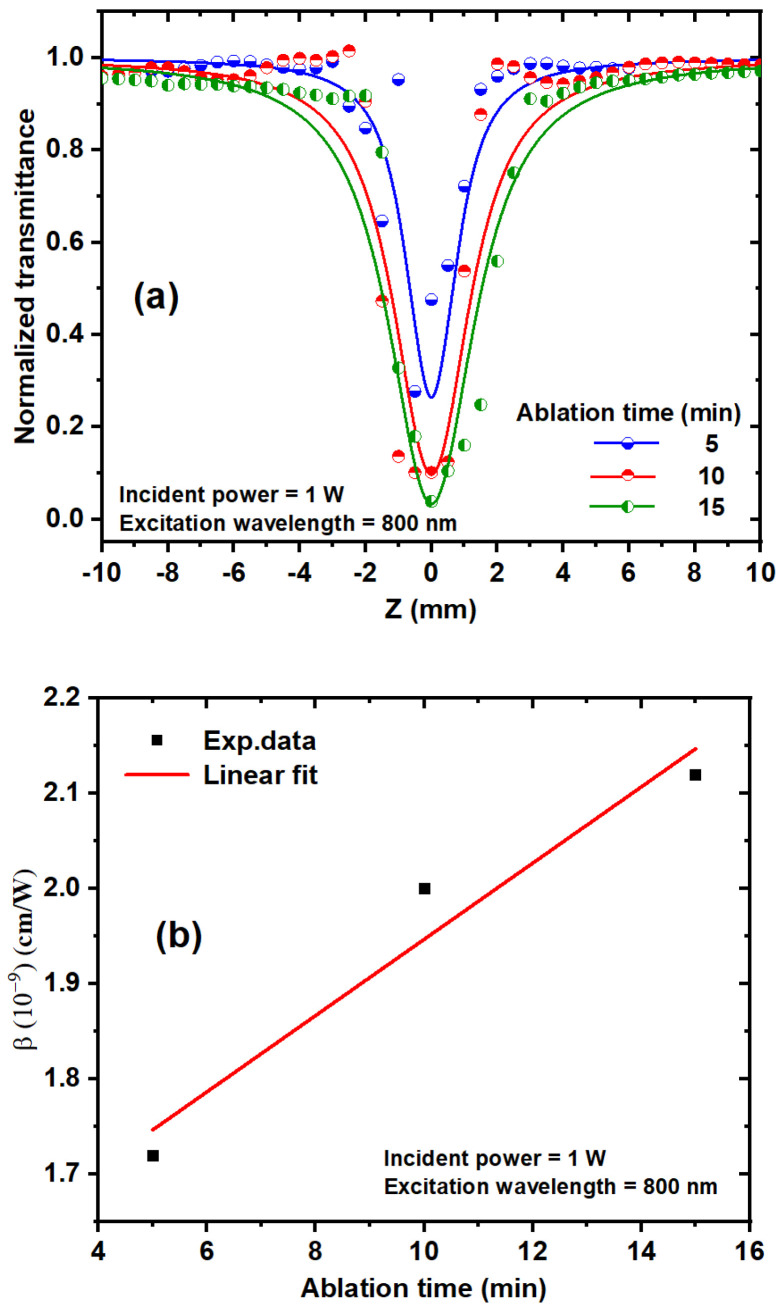
(**a**) OA Z-scan experimental data of TiO_2_ NP colloidal solutions with different ablation times and a constant excitation wavelength of 800 nm and an incident power of 1 W. (**b**) Dependence between the ablation time and the NLA coefficient.

**Figure 10 nanomaterials-14-01940-f010:**
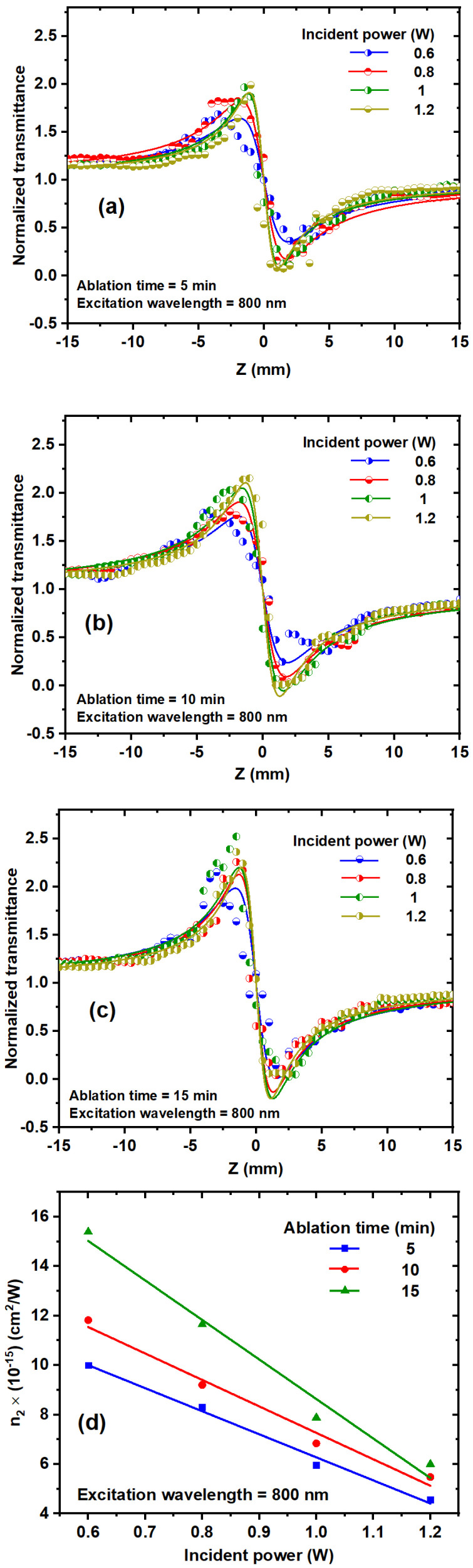
(**a**–**c**) CA Z-scan measurements for TiO_2_ NP colloids at different incident powers and ablation times at an excitation wavelength of 800 nm. The symbols represent the experimental data, and the solid curves are the fits obtained via Equations (6) and (7). (**d**) Relationship between the NLR index and incident power at each ablation time.

**Figure 11 nanomaterials-14-01940-f011:**
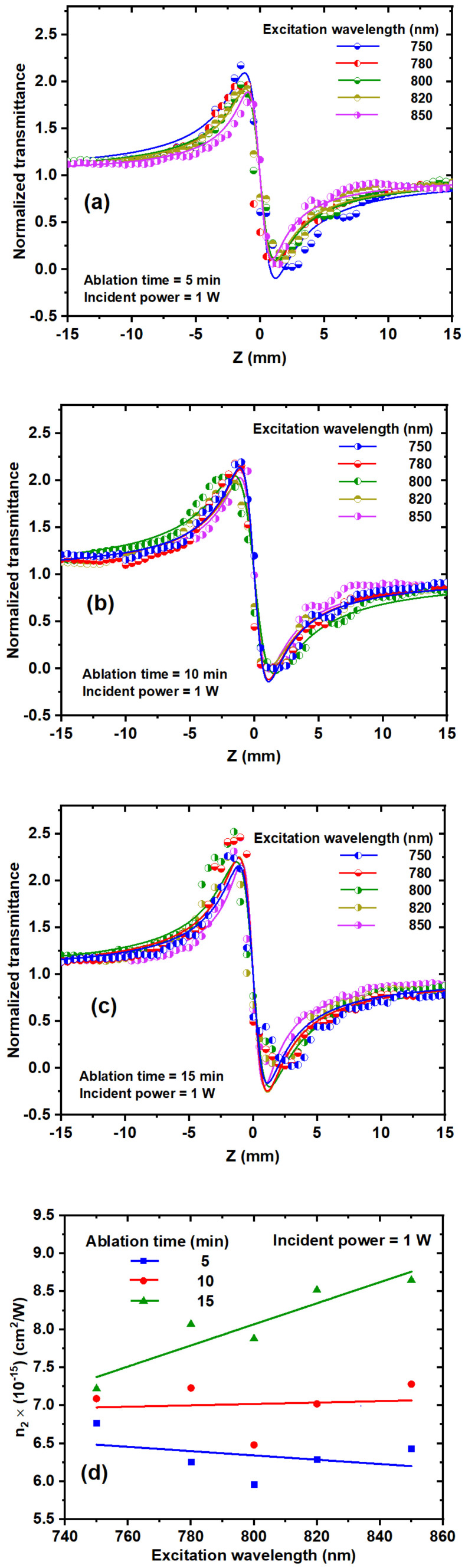
(**a**–**c**) CA Z-scan transmission of TiO_2_ NP colloids at different excitation wavelengths and ablation times; (**d**) relationship values of n_2_ for TiO_2_ NP colloidal solutions with different ablation times at 1 W incident power. The dots represent the experimental data, and the solid curves are linear fits.

**Figure 12 nanomaterials-14-01940-f012:**
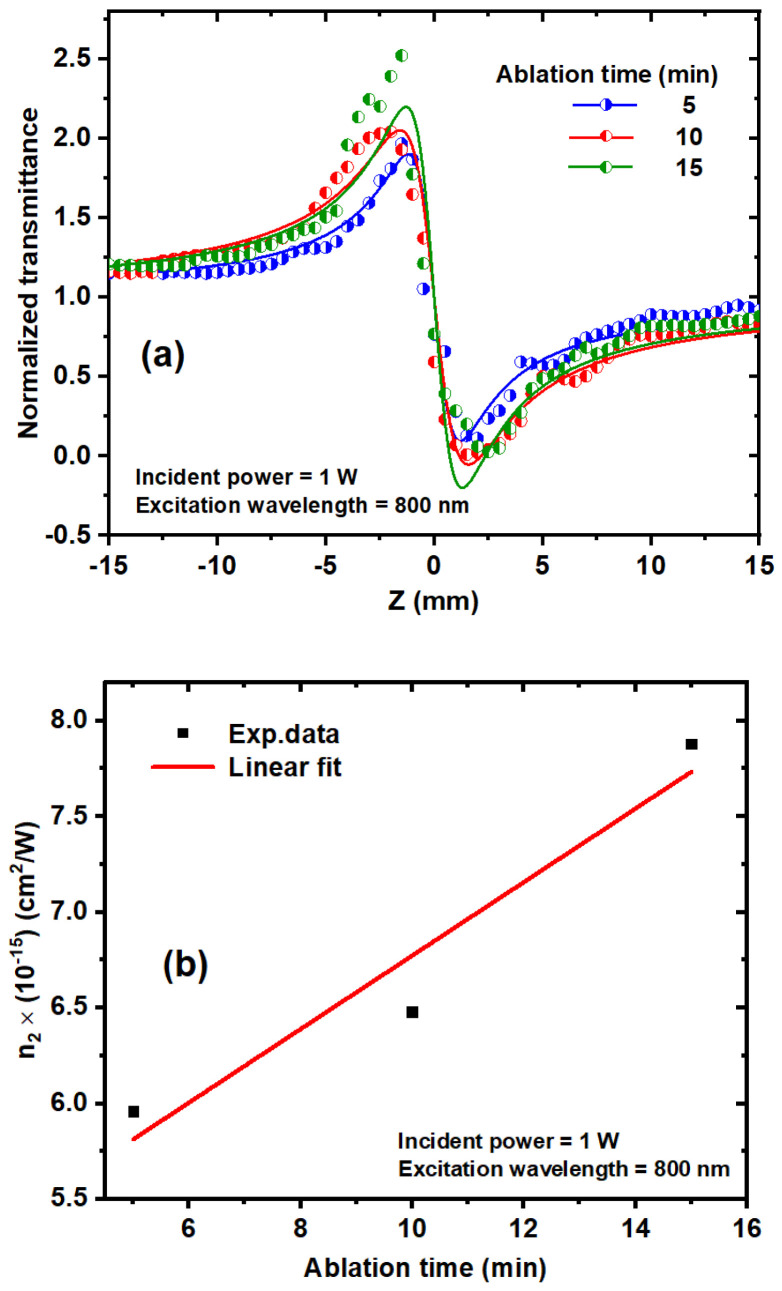
(**a**) Plot of CA Z-scan measurements of different ablation times of 5 min, 10 min, and 15 min at 800 nm excitation wavelength and 1 W incident power. (**b**) Dependence between the measured n_2_ and ablation time at a 1 W incident power and 800 nm excitation wavelength. The dots represent the experimental data, and the solid lines are linear fits.

**Figure 13 nanomaterials-14-01940-f013:**
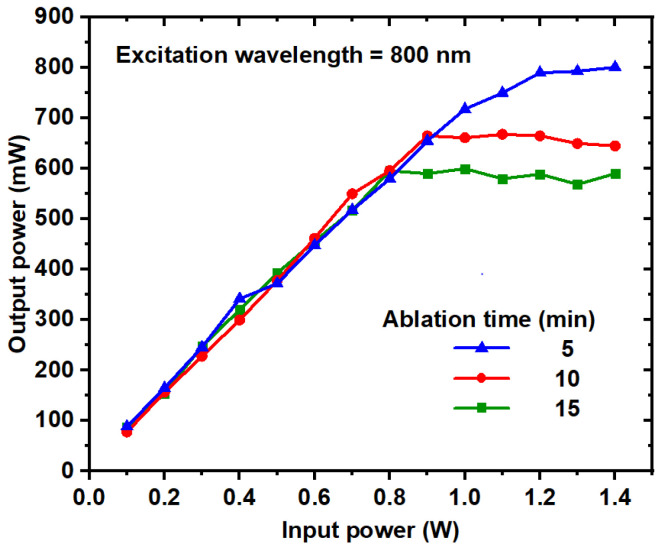
Optical limiting of the TiO_2_ NP colloids at ablation times of 5 min, 10 min, and 15 min and an 800 nm excitation wavelength.

**Table 1 nanomaterials-14-01940-t001:** The linear and nonlinear optical parameters of TiO_2_ NP colloids with different ablation times at an excitation wavelength of 800 nm and an incident power of 1 W.

Ablation Time (min)	TiO_2_ Conc. mg/L	Average Size (nm)	n_0_	α (m^−1^)	n_2_ × 10^−15^ cm^2^/W	ꞵ × 10^−9^ cm/W	$Im\left[\chi^{\left(3\right)}\right]$ × 10^−13^ esu	$Re\left[\chi^{\left(3\right)}\right]$ × 10^−16^ esu	$\chi^{\left(3\right)}$ × 10^−13^ esu	FOM × 10^−11^ esu.cm	Saturation Input Power (W)
5	1.9	19.11	1.35	4.38	5.96	1.72	7.93	8.53	7.93	1.8	1
10	2.9	11.96	1.61	11	6.48	2	13.11	11.07	13.11	1.2	0.9
15	4.35	8.33	1.53	8.7	7.88	2.12	12.55	12.79	12.55	1.44	0.8

**Table 2 nanomaterials-14-01940-t002:** Comparison between the current study and previous studies of TiO_2_ NPs.

Sample	Preparation Method	Phase	Wavelength (nm)	Pulse Duration	Repetition Rate (Hz)	ꞵ (cm/W)	n^2^ (cm^2^/W)	Ref.
Black TiO_2_	Cathodic plasma electrolysis	Film	532	20 ps	1000	−4.9 × 10^−6^	-	[60]
TiO_2_	Chemical	Liquid	532	7 ns	10	2.206 × 10^−8^	3.477× 10^−13^	[61]
TiO_2_	Laser ablation	Liquid	632.8	-	-	0.34 × 10^−3^	−4.3× 10^−8^	[36]
TiO_2_	Chemical	Liquid	1064	7 ns	10	−1.53 × 10^−6^	2.35× 10^−13^	[10]
TiO_2_	Laser ablation	Liquid	800	150 fs	80 × 10^6^	6.2 × 10^−8^	-	[37]
TiO_2_	-	Film	800	50 fs		−6.2 × 10^−9^	−6.32 × 10^−13^	[62]
TiO_2_	Chemical	Liquid	532	7 ns	10	77.8 × 10^−9^	2.9 × 10^−13^	[63]
TiO_2_	Laser ablation	Liquid	800	100 fs	80 × 10^6^	2.12 × 10^−9^	7.88 × 10^−15^	This work

## Data Availability

The data that underlie the results that are presented in this paper are not publicly available at this time but can be obtained from the authors upon reasonable request.

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
