# Peer review of "Experimental Investigation of the Optical Nonlinearity of Laser-Ablated Titanium Dioxide Nanoparticles Using Femtosecond Laser Light Pulses"

_nanomaterials, 2024, doi:10.3390/nano14231940_

Round 1

Reviewer 1 Report

Comments and Suggestions for Authors

The paper by Fatma Abdel Samad et al. reports on the optical nonlinearity of laser ablated TiO2 nanoparticles (NPs) using Z-scan technique.

The motivation of this study may be clarifying the mechanism of the optical nonlinearity of their samples from its excitation power, wavelength, and NPs' concentration dependence. In addtion, they studied the optical limiting behavior and its NPs' concentrantion dependence.

The experimental results are systematic and shows clear tendency against the excitation power, wavelength, and NPs' concentration. However, the reviewer cannot understand their discussion due to the lack of explanations and citation. For example, the discussion about the incident power dependence of the NLA coefficient is in the lenes 230-236. The author listed the components of NLA for explaning the incident power dependence but did not mention which is the main component to be considered here. Judging from the next line, the free carrier absorption may be the main contribution but definition and the discussion of the "many-body interactions" is not in the ref. 50, though ref. 50 is cited here as a reference. The reviewer cannot judge this discussion is acceptable or not.

In addition, there is a serious problem about the structure of the section of the sample characterization. Figure 3 and 4 show the optical spectra of the characterized samples. There are informations about the concentration of the sample. The discussion in the section 3.1 has been done using the information of the concentration and sample size. However these informations are the result in the later sections. The ordering of the sections should be reconsidered.

Moreover, one of the simple question from the reviewer is that why the authors used three kinds of solution sample without changing its concentration. From the result of evaluations, different concentration samples contains different size nanoparticles. To consider the mechanism of the NLA of the NPs, fixing the one parameter must be better. They changed the time of the laser ablation and got the different concentration and particle size sample. These are solution sample with distilled water, then it may be easy to change the concentration by adding the distilled water without changing the particle size. The reviewer thinks this kind of sample preparation can make the result simpler and good for clarifying the mechanism. Is there any reason not to do so?

At present, the reviewer cannot recommend the publication of this paper in Nanomaterials.

Bellow are the minor comments the reviewer noticed during the review process.

1) Caption of table 1 is not good because this table contains the parameter of the linear optical properties.

2) The reviewer cannot understand line 262, "when the laser focused on a typical RSA response". 

3) Ref. 54 may not contain the discussion of dn/dT that is writted in lines 283-285.

4) There may be typo in line 303.

5) The description at line 337 may be wrong especially for Figure 11c.

6) The conclusion in lines 360-362 cannot be understood by the reviewer. There is the change of concentration. Why only the size is the reason?

Author Response

A point-by-point response to Reviewer #1 comments

Dear, 

We appreciate your excellent remarks on our paper, as well as your comments, corrections, and valuable ideas. We believe the following response addresses all of the reviewers' issues. The detailed revisions are listed below where we present the comments of the reviewer followed by our response.

Comment: The experimental results are systematic and show a clear tendency against the excitation power, wavelength, and NPs' concentration. However, the reviewer cannot understand their discussion due to the lack of explanations and citations. For example, the discussion about the incident power dependence of the NLA coefficient is in the lines 230-236. The author listed the components of NLA for explaning the incident power dependence but did not mention which is the main component to be considered here. Judging from the next line, the free carrier absorption may be the main contribution but definition and the discussion of the "many-body interactions" is not in the ref. 50, though ref. 50 is cited here as a reference. The reviewer cannot judge this discussion is acceptable or not.

Response:  Thank you very much for your valuable comments, which we have now modified in the revised version of the manuscript. The number of free electrons in the conduction band (Ti 3d) and holes in the valence band (O 2p) increases as the incident power increases. With increasing incident power, the number of collisions and inelastic scattering between excitons and between excitons and free carriers is so high that they start to interact with each other and alter the NLA coefficient. The collisions of electrons with each other in the conduction band are called "many-body interactions". As a result, collisions between free carriers increase, the scattering of photons and phonons increases, and ꞵ decreases. The NLA coefficient is affected by the change in incident power, taking into account both the average size and concentration of the TiO2 NPs samples.

Comment: In addition, there is a serious problem about the structure of the section of the sample characterization. Figures 3 and 4 show the optical spectra of the characterized samples. There are information about the concentration of the sample. The discussion in the section 3.1 has been done using the information of the concentration and sample size. However, these informations are the result in the later sections. The ordering of the sections should be reconsidered.

Response: Thank you for your valuable comment which we have now incorporated into the revised version of the manuscript. Section 3.1 illustrates the linear optical properties of TiO2 NPs prepared by laser ablation, whereas other sections explain the nonlinear optical properties of TiO2 NPs related to linear parameters.

Comment: Moreover, one of the simple question from the reviewer is that why the authors used three kinds of solution sample without changing its concentration. From the result of evaluations, different concentration samples contains different size nanoparticles. To consider the mechanism of the NLA of the NPs, fixing the one parameter must be better. They changed the time of the laser ablation and got the different concentration and particle size sample. These are solution sample with distilled water, then it may be easy to change the concentration by adding the distilled water without changing the particle size.

Response: We appreciate the reviewer's comments.  That's a great idea, and we intend to carry out this experiment in the future.

Comment: The reviewer thinks this kind of sample preparation can make the result simpler and good for clarifying the mechanism. Is there any reason not to do so?

Response: We appreciate the reviewer's valuable comment. Pulsed laser ablation in liquid (PLAL) enables the synthesis of stable metal colloids in pure solvents without the need for toxic or hazardous chemicals. By adjusting laser parameters such as pulse duration, fluence, repetition rate, wavelength, and pulse width the size of nanoparticles can be precisely controlled, making this method highly versatile and effective.

Comment: Caption of table 1 is not good because this table contains the parameter of the linear optical properties.

Response: We appreciate the reviewer's valuable comment which have been taken into account in the revised version of the manuscript. 

Comment: The reviewer cannot understand line 262, "when the laser focused on a typical RSA response". 

Response:  We appreciate the reviewer comments, which we have now considered in the revised version of the manuscript.

Comment: Ref. 54 may not contain the discussion of dn/dT that is writted in lines 283-285.

Response: We appreciate the reviewer's valuable comment. In the revised  version of the manuscript, the reference 54 has been replaced with a more appropriate one.

Comment: There may be typo in line 303.

Response: We appreciate the reviewer's valuable comment. In the revised  version of the manuscript, the typo has been corrected.

Comment: The description at line 337 may be wrong especially for Figure 11c.

Response: We appreciate the reviewer comments, which we have now considered in the revised version of the manuscript.

Comment: The conclusion in lines 360-362 cannot be understood by the reviewer. There is the change of concentration. Why only the size is the reason?

Response: We appreciate the reviewer's valuable comment which have been taken into account in the revised version of the manuscript. 

Reviewer 2 Report

Comments and Suggestions for Authors

Review for MDPI: Nanomaterials

Title: Experimental Investigation of the Optical Nonlinearity of Laser-Ablated Titanium Dioxide Nanoparticles Using Femtosecond Laser Light Pulses

Authors: Fatma Abdel Samad, Mohamemed Ali Jasim, Alaa Mahmoud, Yasmin Abd El-Salam, Hamza Qayyum, Retna Apsari and Tarek Mohamed

The authors present detailed experimental investigations of the nonlinear optical properties of TiO₂ nanoparticles synthesized through femtosecond laser ablation, specifically employing the Z-scan technique to measure and analyze these properties under varying laser parameters and TiO₂ concentrations. This extensive experimental effort has yielded a detailed characterization leading to a better understanding of the nonlinear properties of TiO₂ nanoparticles. While titanium dioxide nanoparticles have been the subject of extensive research due to their adaptable optical properties, low cost, environmental compatibility, and broad applicability, the study of their nonlinear optical properties when synthesized via femtosecond laser ablation in distilled water remains less prevalent.

The primary subject matter of the article can be classified into the following sections: synthesis and characterization of NP, nonlinear optical measurements, optical limiting, and conclusions. Prior to publication, it is critical to address several issues and provide a more detailed explanation within the manuscript. The most pertinent items are enumerated below:

·         Why were experiments conducted at near-infrared wavelengths when a broadband light source was available? Please provide a brief explanation.

·         Has the occurrence of sedimentation processes involving nanoparticles been observed during the analysis of the sample within the Z-scan system? If so, how should this issue be addressed?

·         The experimental system is described in sufficient detail; however, the estimation of experimental errors in the obtained NLO coefficients at 10% lacks adequate explanation and should be further elaborated.

·         Please clarify how the average size of the TiO2 NPs was determined. As indicated within the insets of Figure 5(a-c), if the scale of the images is maintained, the NPs observable in section (b) appear to be considerably smaller in comparison to those visible in section (c), where the NP size is smaller. This discrepancy is somewhat confusing and requires further clarification.

·         The axis description in Figure 6 is of insufficient size to be readily legible.

·         There is a lack of consistency in the manner of referring to the figures. Please standardize the references to figures, for example: “5(a), (b), and (c)” and “figures 7(a), (b), and (c)”.

·         The content of Paragraph 3.2.2 is limited and lacks sufficient detail. It would be beneficial to expand upon this section and provide a more comprehensive description of figure 8. It would be beneficial to ascertain whether the dependence of the excitation wavelength and NLA coefficient at 1 W incident laser power is indeed linear. In other words, it would be advantageous to determine whether the linear fit observed in Figure 8d accurately represents a fundamental physical phenomenon.

·         The statement that "the NLA coefficient of the TiO2 NPs colloidal solutions is linearly related to the concentration of the NPs, as depicted in Fig. 9 b" (lines 262-263) is a misrepresentation of the data! Given the limited number of data points, it is also plausible that the relationship could be described by a logarithmic or parabolic function. A response to this comment is therefore requested. As illustrated in Figure 11(d), the linear representation also appears to be unsuitable.

·         Optical power limiting. Assuming that the TiO2 NPs colloids effectively attenuate intense laser beam according to the NP concentration, as presented in Figure 13. It would be beneficial to ascertain whether the threshold of such an optical power limiter can be significantly lowered by increasing the NP concentration. Additionally, it would be advantageous to determine if there is any limit above which further increase of NP concentration only leads to a decrease in linear transmission and does not affect power limitation related to nonlinear phenomena. Otherwise, at the presented intensity levels, utilizing such NP in viable real-world devices seems to be of no benefit.

·         Editing errors:
Line 41 – new paragraph
Line 208 – double space

Author Response

A point-by-point response to Reviewer #2 comments

Dear, 

We appreciate your excellent remarks on our paper, as well as your comments, corrections, and valuable ideas. We believe the following response addresses all of the reviewers' issues. The detailed revisions are listed below where we present the comments of the reviewer followed by our response.

Comment: Why were experiments conducted at near-infrared wavelengths when a broadband light source was available? Please provide a brief explanation.

Response: Thank you for your valuable comment. The energy band gap of TiO₂ NPs ranges from 3.06 eV to 2.84 eV at concentrations between 1.9 and 4.35 mg/L. The wavelength range employed for two-photon absorption (2PA) is 750–850 nm, which corresponds to photon energies between 1.65 and 1.458 eV. Nevertheless, the energy of two photons decreases in relation to the energy band gap as the wavelength increases, making two photons insufficient to excite the sample. 

Comment: Has the occurrence of sedimentation processes involving nanoparticles been observed during the analysis of the sample within the Z-scan system? If so, how should this issue be addressed?

Response: Thank you for your valuable comment. We did not notice sedimentation during Z-scan measurements since the TiO₂ NPs colloidal samples were freshly created and characterized in a homogenous colloidal form. This was validated by evaluating the absorption and stability of TiO₂ NPs before experiment. To reduce sedimentation challenges, TiO₂ NPs colloidal samples were continuously stirred before and during data collection.

Comment: The experimental system is described in sufficient detail; however, the estimation of experimental errors in the obtained NLO coefficients at 10% lacks adequate explanation and should be further elaborated.

Response:  We appreciate the reviewer comments, which we have now considered in the revised version of the manuscript.

Comment: Please clarify how the average size of the TiO2 NPs was determined. As indicated within the insets of Figure 5(a-c), if the scale of the images is maintained, the NPs observable in section (b) appear to be considerably smaller in comparison to those visible in section (c), where the NP size is smaller. This discrepancy is somewhat confusing and requires further clarification.

Response: We appreciate the reviewer comments. The average sizes of TiO₂ NPs were determined using high-resolution TEM imaging at various ablation times (5, 10, and 15 minutes), with size distributions calculated from a statistically significant number of particles in each TiO₂ NPs sample. For consistency, all images were captured at the same magnification, and particle sizes were analyzed using calibrated imaging software (Image J). The average size of TiO2 NPs was determined by taking four images for each concentration and at different sample positions.

Comment: The axis description in Figure 6 is of insufficient size to be readily legible.

Response: We appreciate the reviewer comments. The figure has been modified in the revised manuscript.

Comment: There is a lack of consistency in the manner of referring to the figures. Please standardize the references to figures, for example: “5(a), (b), and (c)” and “figures 7(a), (b), and (c)”.

Response: Thank you for your valuable comment, which we have now taken into account in the revised version of the manuscript.

Comment: The content of Paragraph 3.2.2 is limited and lacks sufficient detail. It would be beneficial to expand upon this section and provide a more comprehensive description of figure 8. It would be beneficial to ascertain whether the dependence of the excitation wavelength and NLA coefficient at 1 W incident laser power is indeed linear. In other words, it would be advantageous to determine whether the linear fit observed in Figure 8d accurately represents a fundamental physical phenomenon.

Response: Thank you for your valuable comment, which we have now taken into account in the revised version of the manuscript.

Comment: The statement that "the NLA coefficient of the TiO2 NPs colloidal solutions is linearly related to the concentration of the NPs, as depicted in Fig. 9 b" (lines 262-263) is a misrepresentation of the data! Given the limited number of data points, it is also plausible that the relationship could be described by a logarithmic or parabolic function. A response to this comment is therefore requested. As illustrated in Figure 11(d), the linear representation also appears to be unsuitable.

Response: Thank you for your valuable comment.  We tried to describe the experimental data presented in Figs. 9b and 11d using various forms of fitting, but found that linear fitting was the most appropriate one.

Comment: Optical power limiting. Assuming that the TiO2 NPs colloids effectively attenuate intense laser beam according to the NP concentration, as presented in Figure 13. It would be beneficial to ascertain whether the threshold of such an optical power limiter can be significantly lowered by increasing the NP concentration. Additionally, it would be advantageous to determine if there is any limit above which further increase of NP concentration only leads to a decrease in linear transmission and does not affect power limitation related to nonlinear phenomena. Otherwise, at the presented intensity levels, utilizing such NP in viable real-world devices seems to be of no benefit.

Response: Thank you for your valuable comment.  The optical limiting effect illustrates that at low input powers, the output transmittance's power increases linearly till it reaches a plateau at nearly 0.8 W input power. The data presented in Fig. 13 indicate that the saturation (threshold) input power diminishes with rising TiO2 NPs concentration, as summarized in Table 1. As a result, the outcomes met the criteria necessary for applications on OL devices. The sample is a suitable candidate for nonlinear optical applications, according to all the aforementioned experimental findings.

Comment: Editing errors:
Line 41 – new paragraph
Line 208 – double space

Response: Thank you for your valuable comment, which we have now taken into account in the revised version of the manuscript.

Round 2

Reviewer 1 Report

Comments and Suggestions for Authors

The reviewer thinks the authors almost succeeded in responding the reviewer's comment except for minor points, then would like to recommend to be published in Nanomaterials. However, the reviewer thinks some parts of the manuscript are still confusing because the concentration and the size of the nanoparticles are changed without fixing each of them in this study.

Considering the above, the reviewer would like to recommend some revisions suggested in the following before publication.

1)     The reviewer recommends the authors changing the label of their sample in the text and figures from the concentration to, for example, the duration time in the laser ablation process. This change must be good to avoid the confusion about the sample dependence. In the section 3.1, the authors wrote, “Figure 3 shows how the concentration of TiO2 NPs significantly affects the absorbance spectrum. A surface plasmon resonance (SPR) peak …”. This kind of descriptions should be corrected. The samples in this study cannot be identified by only the concentration of TiO2 NPs because the sample sizes are also changed.

2)     During the revision in this turn, there are several double definition or duplicate expressions in the text. Please correct them. For example,

A)      “Eg is the energy band gap of …” just after eqn. (1) is already abbreviated in the former part.

B)      “The average sizes of TiO2 …” in line 7 on page 6 must be the duplicated description of line 1 on the same page.

C)      The “RSA” in section 3.2.2 is already abbreviated in the former part.

3)     The newly added ref. 50 in the reference list is wrong. Please correct the information.

4)     Almost all the journal name in the References section have “J” on the head. Are these necessary?

5)     Ref. 66 has no journal name. Please check the list of the references again.

Author Response

A point-by-point response to Reviewer #1 comments

Dear, 

We appreciate your excellent remarks on our paper, as well as your comments, corrections, and valuable ideas. We believe the following response addresses all of the reviewers' issues. The detailed revisions are listed below where we present the comments of the reviewer followed by our response.

Comment: The reviewer recommends the authors changing the label of their sample in the text and figures from the concentration to, for example, the duration time in the laser ablation process. This change must be good to avoid the confusion about the sample dependence. In the section 3.1, the authors wrote, “Figure 3 shows how the concentration of TiO2 NPs significantly affects the absorbance spectrum. A surface plasmon resonance (SPR) peak …”. This kind of descriptions should be corrected. The samples in this study cannot be identified by only the concentration of TiO2 NPs because the sample sizes are also changed.

Response:  Thank you very much for your valuable comments, which we have now modified in the revised version of the manuscript.

Comment: During the revision in this turn, there are several double definition or duplicate expressions in the text. Please correct them. For example, A) “Eg is the energy band gap of …” just after eqn. (1) is already abbreviated in the former part. B) “The average sizes of TiO2 …” in line 7 on page 6 must be the duplicated description of line 1 on the same page. C) The “RSA” in section 3.2.2 is already abbreviated in the former part.

Response: Thank you for your valuable comment which we have now incorporated into the revised version of the manuscript.

Comment: The newly added ref. 50 in the reference list is wrong. Please correct the information.

Response: We appreciate the reviewer's comments.  Reference no.50 has been corrected in the revised version of the manuscript.

Comment: Almost all the journal name in the References section have “J” on the head. Are these necessary?

Response: We appreciate the reviewer's valuable comment which have been taken into account in the revised version of the manuscript.  All of the references in the references section have been revised.

Comment: Ref. 66 has no journal name. Please check the list of the references again.

Response: We appreciate the reviewer's valuable comments which have been taken into account in the revised version of the manuscript. 
